# Flux-based hierarchical organization of *Escherichia coli*'s metabolic network

**Semidán Robaina-Estévez**[1,2,3]*, **Zoran Nikoloski**[1,2]

**1** Systems Biology and Mathematical Modeling Group. Max Planck Institute of Molecular Plant Physiology, Potsdam, Germany, **2** Bioinformatics Group, University of Potsdam, Potsdam, Germany, **3** Ronin Institute for Independent Scholarship, Montclair, New Jersey, United States of America

* hello@semidanrobaina.com

**Data Availability Statement:** All data and code files are available from https://github.com/Robaina/fluxOrders (DOI: 10.5281/zenodo.3386033).

## Abstract

Biological networks across scales exhibit hierarchical organization that may constrain network function. Yet, understanding how these hierarchies arise due to the operational constraint of the networks and whether they impose limits to molecular phenotypes remains elusive. Here we show that metabolic networks include a hierarchy of reactions based on a natural flux ordering that holds for every steady state. We find that the hierarchy of reactions is reflected in experimental measurements of transcript, protein and flux levels of *Escherichia coli* under various growth conditions as well as in the catalytic rate constants of the corresponding enzymes. Our findings point at resource partitioning and a fine-tuning of enzyme levels in *E. coli* to respect the constraints imposed by the network structure at steady state. Since reactions in upper layers of the hierarchy impose an upper bound on the flux of the reactions downstream, the hierarchical organization of metabolism due to the flux ordering has direct applications in metabolic engineering.

## Author summary

Metabolism results from the activity of thousands of biochemical reactions, which create, transform and recycle all chemicals, i.e., metabolites, required by life. Metabolic reactions depend on enzymes—proteins acting as biological catalyzers—to proceed, which effectively links metabolism to other layers in the organization of cellular physiology, such as transcription and translation. These reactions do not operate in isolation but interact in a metabolic network due to the metabolites they share. The joint action of the reactions imposes constraints on the reaction fluxes denoting rates of conversion between metabolites. Here, we identify one such constraint imposed by the metabolic network of *E. coli*, which we call the flux order relation. Specifically, we identify pairs of reactions in which one reaction always carries a higher flux than the other at steady state. Hence, the flux order relation creates a hierarchical organization of metabolism. We show that the flux order relation is reflected in experimental data sets of fluxes, gene expression, protein levels, and enzyme catalytic constants. Our results point at resource partitioning and a fine-tuning of enzyme levels in *E. coli* to respect the flux order relation.

**Funding:** The authors received no specific funding for this work.

**Competing interests:** The authors have declared that no competing interests exist.

## Introduction

Hierarchical organization has been recognized as a salient feature of biological networks, spanning from the molecular to the ecosystem scale [1–6]. Biological networks do not operate in isolation but form interconnected layers. It is therefore important to identify the constraints that a hierarchical organization in one cellular layer impose on the others. For instance, metabolism is understood as an integrated outcome of biochemical reactions and transcriptional, translational, and post-translational changes with input from the environment. Therefore, elucidating metabolic hierarchies can contribute to understanding the constraints governing physiology as well as their implications for biotechnological applications. Two types of metabolic hierarchies have been identified: The first is based on nested subnetworks, while the second relies on pairwise relationships between components. Nested subnetworks can arise solely due to structural properties of the components in the network. For instance, it has been shown that metabolic networks can be decomposed into modules of highly connected metabolites, and that these modules form a nested hierarchy [5, 7]. Further, a nested hierarchy of bow-tie structures, in which a few intermediate metabolites transform a large number of input and output metabolites, has also been identified in metabolic networks [8]. However, additional hierarchies of nested subnetworks arise when operational principles, like mass balance and steady-state constraints or optimality of cellular functions, are considered. Such hierarchies include the so-called self-maintained subnetworks (e.g. chemical organizations [9, 10]) and subnetworks that operate as a unit when the network optimizes a cellular objective (e.g. flux-modules [11–13] and feasible coalitions [14]).

Moreover, hierarchies that rely on pairwise relationships between components often consider operational principles in conjunction with the network structure. For instance, mass-balance and steady state constraints also induce hierarchies based on asymmetric, pairwise relations among reactions (Fig 1c). One such relation is the so-called directional flux coupling relation, in which one reaction controls the activation state of another in any steady state of the network [15]. The directionality (i.e. asymmetry) of this coupling relation induces a hierarchical organization, in which reactions situated at the top of the hierarchy control the activation state of all reactions accessible below [16].

Several studies have already revealed the constraints of metabolic network structure and steady-state operation to the levels of associated genes and proteins. For instance, genes associated to stoichiometrically coupled reaction pairs tend to be co-expressed in *E. coli* [17] and maize [18] and to co-evolve in *E. coli* [19]. Further, genes associated to reactions in upper layers of the hierarchy based on directional flux coupling tend to be more regulated at the transcriptional level [16]. In addition, computational and experimental evidence has indicated that reactions in the core of a metabolic network (e.g. the citric acid cycle) generally carry higher flux than reactions in the network periphery [20], that flux distributions resemble a power law distribution in the case of *E. coli* [21] and that genes associated to the reactions carrying higher flux tend to show decreased evolutionary rates [20]. Yet, it is unclear whether there exists a hierarchy based on flux levels, whereby fluxes of reactions at the top would impose an upper bound to the fluxes of the reactions below in the hierarchy. Moreover, the extent to which such hierarchy would affect the levels of upstream components and their biochemical properties is unclear.

Here, we define the flux order relation and characterize the flux order hierarchy found in *E. coli*'s metabolic network. Further, we investigate the effects of the flux-based hierarchy on upstream components in the cellular organization. To this end, we use data from *E. coli* to evaluate if and to what extent the hierarchy induced by the flux order relation is manifested in transcript levels, protein abundances (for growth under three different carbon sources),

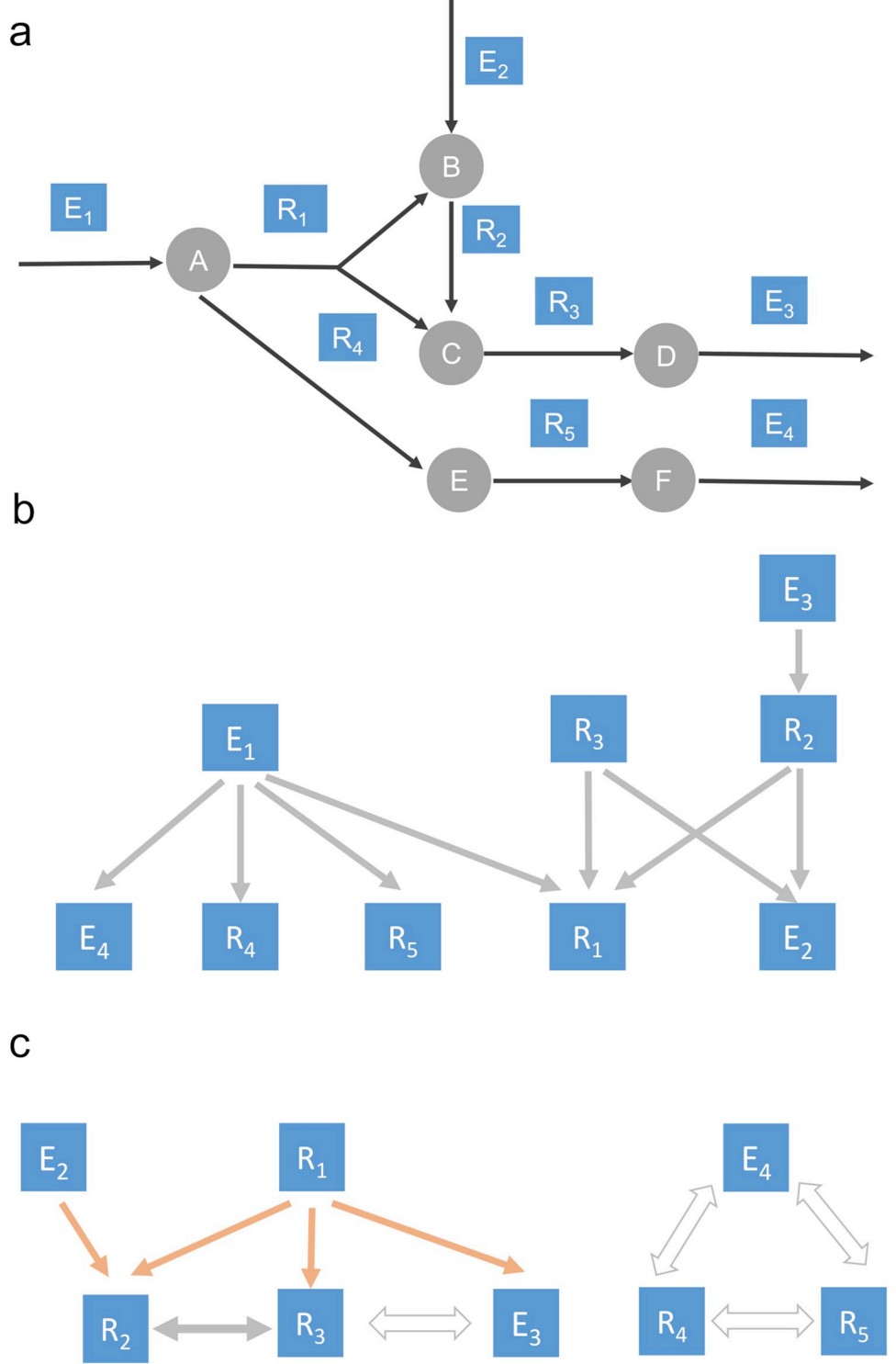

**Fig 1. Illustration of the flux order relation and comparison to flux coupling relations.** (a) In this toy metabolic network, reactions are depicted as arrows and named in the blue squares while metabolites are depicted as circles. Internal reactions are denoted with R, while exchange reactions with E. (b) Flux order graph corresponding to the toy metabolic network depicted in (a), here, a reaction is connected to another by a directional edge if it carries a greater of equal flux in any steady state. (c) Flux couplings graph for the toy metabolic network depicted in (a), directionally coupled relations are depicted as orange arrows, while partially and fully coupled relations are illustrated with double small and large arrows, respectively. Note that the directionally coupled relation is the only one that induces a directed acyclic graph (see main text).

measured fluxes (for growth on glucose) and enzyme catalytic constants. By comparative analysis with coupling relations, we show that the flux order relation induces a hierarchy in *E. coli*'s metabolic network. Finally, we demonstrate that the flux order relation pinpoints reactions limiting the flux through target processes, which has direct biotechnological applications.

## Methods

In this section, we provide a general description of the computational methods employed in this study. For a more detailed description, as well as to fully reproduce the results presented in this study, we refer the reader to the GitHub repository https://robaina.github.io/fluxOrders containing a complete workflow of the study as well as all the code and data employed.

### Background ad definitions

A biochemical network is represented by the stoichiometric matrix, $S$, in which columns represent reactions and rows metabolites. Each entry $s_{ij}$ in $S$ corresponds to the stoichiometry of metabolite $m_i$ in reaction $r_j$: $s_{ij} < 0$, when the species is a substrate, $s_{ij} > 0$ when it is a product, or $s_{ij} = 0$, when is not involved in the reaction. In this study, we split reversible reactions into the forward and backward direction, hence the reaction flux is always non-negative. The rate of change of the concentration $x_i$ of metabolite $m_i$ is then described as

$$\frac{d}{dt} x_i = \sum_{j=1}^{n} s_{ij} v_j. \tag{1}$$

When the system 1 is in steady state, i.e., metabolite concentrations are constant over time, we have $Sv = 0$, which forms a set of mass balance linear equations. The set of mass balance linear equations plus the constraint $v_j \geq 0$, define a polyhedral cone $C = \{v: Sv = 0, v \geq 0\}$, containing all valid flux vectors, i.e., flux distributions at steady state.

We consider a reaction pair $r_i$, $r_j$ ordered if $v_i \geq v_j$, $\forall v \in C$. The flux order relation satisfies the three properties of a partial order relation: *(i)* it is reflexive, i.e., $v_i \geq v_i$, *(ii)* antisymmetric, i.e., $v_i \geq v_j \wedge v_j \geq v_i \Rightarrow v_i = v_j$ and *(iii)* transitive, i.e, $v_i \geq v_j \wedge v_j \geq v_k \Rightarrow v_i \geq v_k$. Therefore, reactions can be part of ordered chains in which $v_i \geq v_j \geq v_k$.

### Identification of flux-ordered reaction pairs

We look for pairs of reactions $r_i$, $r_j$ which satisfy the flux order relation, i.e. $\forall v_i$, $v_j \in \{v: Sv = 0, v_{min} \leq v \leq v_{max}\}$, $v_i \geq v_j$. To this end, we can evaluate the ratio of fluxes in the feasible region, i.e., we need $v_i/v_j \geq 1$. We can evaluate the ratio with the following linear-fractional program:

$$\rho_{min}, \ \rho_{max} = \min_v, \max_v \ \frac{v_i}{v_j}$$

$$\text{s.t.}$$

$$Sv = 0 \tag{2}$$

$$v_{min} \leq v \leq v_{max}$$

Then $v_i \geq v_j \Leftrightarrow \rho_{min} \geq 1, \rho_{max} > 1$. However, linear-fractional programs cannot be directly solved with common linear programming tools. Fortunately, the Charnes-Cooper

transformation [22] allows us to express the linear program (LP) in Eq 2 as

$$z_{min}, \; z_{max} = \min_{w,t}, \max_{w,t} w_i$$

s.t.

$$Sw = 0$$

$$w_j = 1 \qquad\qquad (3)$$

$$tv_{min} \leq w \leq tv_{max}$$

$$t \geq 0$$

in which $w_i = v_i/v_j$. We could then solve the two LPs in Eq 3 for each pair $r_i$, $r_j$ or reactions and evaluate whether $z_{min} \geq 1, z_{max} > 1$ is satisfied, which implies that our original flux pairs satisfy $v_i \geq v_j$. However, due to the large number of LPs required to be solved—i.e., $n(n-1)$ for a model with $n$ reactions—we implemented a pre-processing step to discard reaction pairs that cannot be flux-ordered.

## Identification of candidate flux-ordered pairs

The pre-processing step was conducted as follows. A random sample of flux distributions—vectors of flux values—at steady state and under the relevant environmental conditions was first obtained using the *cobrapy* library [23]. Then, reaction pairs, $r_i$, $r_j$ for which $v_i \geq v_j$ hold in the entire sample were further evaluated (here named as candidate ordered pairs) and those which showed an inconsistent order relation throughout the sample were removed. The order relation was considered inconsistent if $v_i \geq v_j$ and $v_i \leq v_j$ were both true in at least two sampled flux distributions. Further, we removed fully coupled reaction pairs from the set of candidate flux ordered pairs. Reaction pairs that show a fully coupled relation in their fluxes satisfy $v_i = \alpha v_j$ for a constant $\alpha \in \mathbb{R}$ at any steady state [15], hence the order relation can be inferred directly from $\alpha$ once a flux distribution at steady-state is found *e.g.*, through flux balance analysis. Fully coupled reaction pairs were identified using the F2C2 toolbox [24], more details in section: Analysis of flux coupled reaction pairs.

The LP 3 was solved by the *Gurobi* solver, version 8 (https://www.gurobi.com) under the R interface. The iteration over candidate ordered reaction pairs was run in parallel. To this end, the following R libraries were employed: *Matrix, slam, forEach, parallel* and *doParallel*.

## Analysis of flux coupled reaction pairs

We employed the F2C2 toolbox [24], which is implemented in MATLAB, to identify flux coupling relations among reaction pairs. We slightly modified the original F2C2 code to switch off the pre-processing step, in which blocked reactions are removed. This modification allowed us to avoid numerical discrepancies between MATLAB's F2C2 code and the native *cobrapy* function to find blocked reactions. Thus, blocked reactions were only identified once using *cobrapy*, and then the reduced model without blocked reactions was passed to the F2C2 toolbox.

## Reconstruction of the flux order DAG

The flux order relation induces a partially ordered set (poset) on the set of metabolic reactions of the genome scale model. The poset contains the hierarchical structure of the ordered reaction chains and can be represented as a directed acyclic graph (DAG). In the DAG, nodes correspond to elements of the set, metabolic reactions in this case, and edges represent the order

relation. More specifically, an edge drawn from node $v_i$ to $v_j$ indicates that $v_i \geq v_j$. The transitive reduction of a DAG is another DAG in which only the edges linking nodes in a (transitive) ordered chain are maintained. The transitive reduction of the DAG is also termed the Hasse diagram of the poset, although in this study we employ the term flux order DAG. For instance, in the chain $v_i \geq v_j \geq v_k$, the DAG would contain the edges $v_i \rightarrow v_j$, $v_j \rightarrow v_k$ and also $v_i \rightarrow v_k$. However, the last edge is implied by transitivity and hence it is redundant: there is already a path between $v_i$ and $v_k$, namely $v_i \rightarrow v_j \rightarrow v_k$.

We employed the python module *networkx* [25] to reconstruct a DAG from the adjacency matrix representing the flux order relation among reaction pairs. In the adjacency matrix, entries $A_{i,j} = 1$ represent the order relation $v_i \geq v_j$, while $A_{i,j} = 0$ represents an unordered reaction pair. Fully coupled reaction pairs with equal flux values, i.e., with $\alpha = 1$, and by transitivity, all chains of fully coupled reactions with equal flux were collapsed into a single node when reconstructing the flux order DAG. The transitive reduction of the DAG was then computed with *networkx* to obtain the final DAG that we employed in our analyses.

## Extracting levels in the flux order DAG

To classify reactions into the different hierarchical levels of the flux order DAG, we employed a similar approach to the one conducted by Hosseini et al. [16]. Specifically, for each reaction in the DAG, we defined its level as the longest shortest path between the reaction and any of the root reactions, i.e., reactions in the first level. To this end, we employed the python library *networkx* [25] to compute the longest shortest path.

## Model, growth media and data preparation

The genome-scale metabolic model iJO1366, corresponding to the strain K-12 sub-strain MG1566 of *E. coli* was downloaded from the BIGG database [26] and imported into python with the *cobrapy* library [23]. Blocked reactions, i.e., reactions that cannot carry flux at steady state, were removed from the model, and reversible reactions were split into the forward and backward reaction. Thus, the final model was composed of irreversible non-blocked reactions.

We simulated three different aerobic minimal media containing a single carbon source: glucose, glycerate or acetate. To this end, we only allowed the import of oxygen, inorganic compounds and the sole carbon source for each medium. We set the maximum import rate for each carbon source to be 20 mmol.DW.h$^{-1}$, while the import rate of the rest of imported compounds was unconstrained. The minimum and maximum flux bounds of internal reactions were set to the default values of iJO1366. These include a minimum flux through the non-growth-associated maintenance reaction *ATPM* of 3.15 mmol.DW.h$^{-1}$. Additionally, we constrained the minimum flux through the biomass reaction *BIOMASS_Ec_iJO1366_WT_53p95M* to be 95% of the optimal biomass production rate under the established conditions.

All data were obtained from a collection of published studies and, in all cases, data corresponded to the wild type of *E. coli*'s K-12 strain measured under no environmental stresses. Transcript and flux levels were obtained from a compendium of several studies [27]. While transcript levels were measured under the three carbon sources analysed in this study, flux level data were only available for growth in minimal medium with glucose as sole carbon source. Protein levels and $k_{cat}$ values were obtained from Davidi et al. (Supplementary Dataset 1) [28]. The used $k_{cat}$ values were obtained from the BRENDA database [29] and are manually curated by Davidi et al. [28]. We obtained protein metabolic cost data from Kaleta et al. (Supporting data S1) [30]. Protein cost data were measured as moles of carbon source (data

available for glucose and acetate) consumed per mole of protein synthetized. In the case of glycerate, we employed the protein costs measured under glucose.

## Analysis of order relations in data

We applied the following strategy to evaluate to what extent the flux order relation was respected by experimental measurements. We first assigned data values to every reaction in the iJO1366 model. In the case of transcript and protein data, we collected the data corresponding to all genes coding for the enzyme—displayed in the Gene-Protein-Reaction rules included in iJO1366—across all replicates for the selected carbon source. In the case of flux levels and $k_{cat}$ values, we simply collected all available data for each reaction. In this case, only data for glucose as carbon source were available. Altogether, this procedure generated a distribution of data values for each reaction and carbon source.

We next tested whether data values assigned to two ordered reactions tended to follow the order relation. To this end, we computed the average difference between any two pairs of data values from the two distributions: If a reaction pair $r_i$, $r_j$ with data values $D_i = \{d_i^1, d_i^2, \ldots, d_i^n\}, D_j = \{d_j^1, d_j^2, \ldots, d_j^n\}$ satisfied $v_i \geq v_j$ in any steady state, we computed

$$\hat{\delta}_{ij} = \frac{1}{nm} \sum_{p=1}^{n} \sum_{q=1}^{m} (d_i^p - d_j^q)$$

or equivalently $\hat{\delta}_{ij} = \hat{D}_i - \hat{D}_j$ where $\hat{D}$ corresponds to the mean data value assigned to each reaction. After applying this procedure to all ordered pairs of reactions, we obtained $\Delta = \{\hat{\delta}_{ij} \; \forall (i,j) : v_i \geq v_j\}$, the distribution of average differences between data values of ordered pairs. We considered the fraction of positive average difference of data values, i.e., $f_{>0} = \Pr(\delta_{ij} > 0)$, as an indicator of the tendency of data to follow the flux order relation. We considered the positive fraction instead of the nonnegative fraction, i.e., $f_{\geq 0} = \Pr(\delta_{ij} \geq 0)$, to account for the possible cases in which two ordered reactions were assigned the exact same set of data values, thus producing a mean data difference of exactly zero and artificially increasing the proportion of matches in our analysis. This scenario could happen when the two enzymes share the same set of genes for which data are available.

We further performed a permutation analysis to test whether $\Delta$ was significantly different from a distribution of random average data differences. To this end, we first randomly assigned data values to each gene, in the case of transcript and protein data, or reaction, in the case of flux data and $k_{cat}$ values. The data values were drawn from the set of all available data values associated to the genes or reactions in the iJO1366 model. We then computed the distribution of mean differences across all ordered reaction pairs with available data and repeated this process $n = 10000$ times, obtaining the fraction of positive average differences in each case. We computed the empirical $p$-value $= (r + 1)/(n + 1)$, where $r$ is the number of random samples in which the fraction of positive average differences was greater than or equal to that obtained with the original dataset.

## Flux orders in ¹³C-measured fluxes

The flux data employed in this study were retrieved from the Ecomics database [27], which, in turn, contained four different sources of flux data [31–34]. However, 32 out of the total 43 flux data samples in Ecomics came from Ishii et al. [34]. Hence, we used the core metabolic network employed by Ishii et al. [34] to determine their measured flux data as our experimental network. We reconstructed a stoichiometry matrix from the reaction information displayed in supplementary SFigure 2 and supplementary tables S4 and S5 of Ishii et al. [34], and set a

maximum capacity of 1000 mmol.DW.h$^{-1}$ to each reaction in the network except the uptake of glucose, which had a maximum capacity of 20 mmol.DW.h$^{-1}$.

We then proceeded to find the ordered reaction pairs following the same workflow employed in the case of the iJO1366 model. Reaction identifiers employed in the flux database were mapped to iJO166 reaction names following the dictionary in the file supplementary data 6 of reference [27]. The generated experimental reaction network can be found in S2 Table of this study.

## Gene regulation in the flux order and flux coupling DAGs

To reconstruct the flux-coupling DAG, we first obtained all flux coupled pairs of the iJO1366 model with the F2C2 tool (see section: Analysis of flux coupled reaction pairs). Next, we selected all directionally coupled pairs and discarded fully and partially coupled pairs since these cannot form a hierarchy (they are symmetric relations). We obtained in this manner an adjacency matrix which we employed to reconstruct the flux coupling DAG, similar to the case of the flux order DAG (see section: Reconstruction of the flux order DAG).

We obtained the number of regulatory interactions for the genes in iJO1366 with available data from the RegulonDB database [35]. In particular, we retrieved the total number of regulatory interactions, i.e., including both inhibitory and activating. Next, we employed the Gene-Protein-Reaction rules in the iJO1366 model to determine which genes coded for the enzymes catalysing the reactions in each graph level. In addition, we pooled all data values to form a distribution of the total number of regulatory interactions per graph level.

We partitioned the set of levels of the DAGs into three subsets, termed: first, middle and last, to obtain distributions that contained a similar amount of data values. (The first levels in the DAG contained the majority of reactions in both the flux order and the flux coupling DAG.) Specifically, the first subset contained reactions in the first level of the DAGs, the middle, reactions in the second level and the last, reactions in all remaining levels of the DAGs. To evaluate the statistical significance of the comparisons between levels, we run a one-sided Mann-Whitney U test, with $\alpha = 0.05$, between the first and middle, first and last, and middle and last subsets of levels. To this end, we employed the Mann-Whitney implementation of the *scipy* python library.

## The flux order relation in the context of reaction essentiality

To reconstruct the DAG of reactions that had a greater of equal flux value than the biomass reaction, we extracted the subgraph composed of all ancestors of the biomass reaction in the complete flux order DAG (for glucose as carbon source) using the *networkx* library in python [25]. Additionally, while the ATP maintenance reaction (ATPM in iJO1366) was an ancestor of the biomass reaction in the complete flux order DAG, we removed it from the essentiality subgraph. In fact, the ATPM reaction was artificially flux-ordered with the biomass reaction, since we constrained the lower flux bound of ATPM to a value (see Methods, section: Model, growth media and data preparation) that was always greater than the maximum possible biomass production value for the unconstrained iJO1366 model. The resulting (sub)DAG was visualized with *cytoscape 3* [36]. Essential reactions were also identified with *cobrapy*'s method *find_essential_reactions* [23]. In this manner, we could find the set of reactions that were both flux-ordered with the biomass reaction and essential as well as the set of reactions that were essential but not flux-ordered.

## Results & discussion

We investigated the flux order relation in the iJO1366 model, a genome-scale metabolic reconstruction of *E. coli*'s K-12 strain [37], and simulated growth in minimal medium under three different carbon sources: glucose, acetate and glycerate (Methods, section: Model, growth media and data preparation). First, we characterized the flux order relation in the iJO1366 model for the three carbon sources by using techniques from constraint-based metabolic modelling and mathematical optimisation (Methods, section: Background ad definitions). Next, we analysed the occurrence of metabolic pathways in iJO1366 along the hierarchy. Finally, we evaluated whether the flux order relation was respected by diverse experimental data sets (Methods, section: Analysis of order relations in data). An online Jupyter Notebook containing the workflow followed in this study as well as additional interactive figures can be accessed from https://robaina.github.io/fluxOrders.

### Background ad definitions

We followed a constraint-based approach to model the operation of the metabolic network at steady state [38]. In constraint-based metabolic modelling, the space of all feasible distributions of reaction fluxes $v$ at steady state is determined by the linear system $Sv = 0$, and the flux bound constraints $v_{min} \leq v \leq v_{max}$, in which $S$, the stoichiometric matrix, captures the structure of the metabolic network (Methods, section: Background ad definitions, Fig 1a). We consider a pair of reactions to be flux-ordered if their fluxes satisfy $v_i \geq v_j$ in any steady state of the system (Methods, section: Background ad definitions).

The flux order relation can be represented by a directed acyclic graph (DAG), which we call here the flux order DAG (Methods, section: Reconstruction of the flux order DAG). For instance, Fig 1a displays a toy metabolic network composed of nine reactions and five metabolites. In this example, reactions $E_3$ and $R_2$ are flux-ordered, i.e., $v_{E_3} \geq v_{R_2}$ in any steady state, which we denote as $E_3 \geq R_2$. We represent the order relation in the DAG in Fig 1b, where nodes depict reactions and directed edges the flux order relation. Hence, an edge connects reaction $E_3$ to reaction $R_2$. Due to the transitivity of the flux order relation (Methods, section: Background ad definitions), the flux order DAG contains chains of reactions which are all flux-ordered and in which the first reaction (a root) in the chain imposes and upper flux bound to the remaining reactions. In the example of Fig 1b, we have that $R_2 \geq E_2$ and by transitivity $E_3 \geq E_2$. The triad $E_3, R_2, E_2$ forms a flux-ordered chain, $E_3 \geq R_2 \geq E_2$, in which $E_3$ is the root. See Section 2.3 of the online Jupyter Notebook for an illustration of a flux-ordered chain in the iJO1366 model. Additionally, we provide a web application to explore the core section of the flux order DAG of iJO1366 in https://robaina.github.io/fluxOrders.

### The flux order relation in *E. coli*

We identified the flux-ordered reaction pairs for the three carbon sources: glucose, acetate and glycerate in an scenario of aerobic growth, i.e., we constrained the minimum flux through the biomass reaction to be 95% of the maximum possible (Methods, section: Identification of flux-ordered reaction pairs).The total number of flux-ordered reaction pairs was similar among the three carbon sources, with an average number of 119,019.3, i.e., 5.4% of the total number of reaction pairs in iJO1366 (online Jupyter Notebook, Section 1.3). However, there were 94,071 shared flux-ordered pairs among the three carbon sources, which represented as much as 91.8% of all flux-ordered pairs for glucose, and lower values of 75.33% for glycerate, and 72.49% for acetate. In fact, acetate imposed the largest number of 12,382 (9.54%) exclusive flux-ordered pairs (online Jupyter Notebook, Section 1.3). Further, the pairwise comparison,

employing the Jaccard distance, indicated that acetate and glycerate shared more flux-ordered pairs in comparison to glucose (online Jupyter Notebook, Section 1.3). These findings show that flux-ordered pairs are condition-dependent and allow us to distinguish functionality under different carbon sources.

## A comparison between flux ordering and flux coupling

Next, we asked if the flux order relation differed from flux coupling relations, which have already been used to discover hierarchies in metabolic networks [16]. To this end, we compared the flux order relation with the three types of flux coupling: full, partial and directional [15]. Flux coupling relations impose constraints on the activity of the reactions and on the ratio of their flux values. Specifically, for a pair of reactions, both full and partial coupling relations impose a bidirectional matching of the reaction activity, i.e. when one of the reactions carries non-zero flux, the other one must also carry non-zero flux (and vice versa). The fully coupled relation implies, in addition, that the ratio of the two fluxes is constant in every steady state, i.e., $v_i/v_j = \alpha$, for $\alpha \in \mathbb{R}$, while the partially coupled relation requires a positive, finite bound on the ratio i.e., $0 < v_i/v_j \leq \beta$, with $\beta > 0$ being a finite upper bound for the entire space of steady states. In contrast, the directional coupling relation only imposes a unidirectional match of the reaction activity, i.e., one (leading) reaction carrying non-zero flux causes another reaction to carry non-zero flux in any steady state.

Given the constraints that the flux coupling relations impose in the flux ratio of certain reaction pairs, we investigated whether the flux-ordered pairs were fully represented by the flux coupling relations. First, we found that the total number of flux-ordered pairs was larger than that of coupled pairs. Specifically, while the average number of flux-ordered pairs for the three carbon sources was 119,019.3 (5.4%) (online Jupyter Notebook, Section 1.3), the number of all three types of flux-coupled pairs was 42114 (1.9%). Next, we evaluated the intersection of flux-ordered pairs with the flux-coupled pairs. We found that an average of 22.39% of the flux-ordered pairs over the three carbon sources was also flux-coupled. The largest fraction of flux-coupled pairs among the flux-ordered was found for glucose (25.23%), followed by glycerate (21.31%) and acetate (20.61%). Moreover, the majority (94.4%) of coupled and ordered pairs were directionally coupled (Table 1). Finally, we evaluated the fraction of partially and directionally coupled pairs that were also ordered. Here we did not consider the fully coupled pairs since they are ordered by definition. Across the three carbon sources, we found that an average of 81.7% of the directionally and of 92.7% of the partially coupled were also flux-ordered (Table 1).

Altogether, these results show that while most flux coupled pairs are also flux-ordered, the flux order relation is considerably richer than the three flux coupling relations, and thus can provide novel insights in the hierarchical organization of metabolism.

**Table 1. Overlap between flux-ordered and flux-coupled reaction pairs.** Conditional probabilities between the sets of flux-ordered reaction pairs (O), the union of all three types of flux-coupled reaction pairs (C), directionally coupled (D), partially coupled (P) and fully coupled (F) pairs across the three carbon sources evaluated in this study. A majority of flux-ordered reaction pairs are not flux-coupled while most flux-coupled pairs are also flux-ordered, thus the flux order relation is not fully represented in all three coupling relations.

| | Pr(C \| O) | Pr(D\|O) | Pr(P \| O) | Pr(F \| O) | Pr(O \| C) | Pr(O \| D) | Pr(O \| P) |
|---|---|---|---|---|---|---|---|
| Glucose | 0.3374 | 0.3187 | 0.0171 | 0.0017 | 0.8158 | 0.8314 | 0.9278 |
| Glycerate | 0.2710 | 0.2557 | 0.0140 | 0.0014 | 0.7990 | 0.8134 | 0.9252 |
| Acetate | 0.2597 | 0.2450 | 0.0134 | 0.0013 | 0.7955 | 0.8098 | 0.9236 |

## Properties of the hierarchy implied by the flux order relation

We next investigated how reactions and metabolic (sub)systems were distributed across the hierarchy implied by the flux order DAG. To this end, we first classified reactions in the DAG into different levels according to their distance to a root reaction. Specifically, we defined the level of a reaction as the number of reactions in the shortest chain to a root. Hence, the larger the level of a reaction, the more constrained the largest flux value it may take in any steady state (Methods, section: Extracting levels in the flux order DAG).

After reconstructing the flux order DAG for the three carbon sources, we found a maximum number of 20 levels for glucose and glycerate, while the DAG for acetate contained one additional level. Further, the majority of reactions were located in the first levels of the DAG in the three carbon sources, specifically, the first six levels contained over 90% of the reactions (Fig 2a). The number of sources, i.e., reactions in the first level (i.e., level 0 in Fig 2a), was markedly greater than the number of sinks, i.e., reactions that do not carry a flux value greater than any other reaction in every steady state. Therefore, in line with observations of purely structural hierarchy in gene regulatory networks, the hierarchies based on flux orders for all three carbon sources is branching, but relatively shallow [39].

Next, we asked how metabolic pathways and systems were distributed across the levels of the DAG. The iJO1366 model contains a total of 32 metabolic subsystems. To facilitate the analysis, we grouped the subsystems into eight larger categories, here termed macrosystems (S1 Table). We first looked at the distribution of macrosystems among all reactions present in the DAGs of all three carbon sources as well as the sets of reactions which were exclusive to each carbon source. We found that among the set of shared reactions, *Transport* was the macrosystem with the largest number of reactions, followed by *Amino acid* and *Lipid metabolism*. However, *Transport* was not the dominant macrosystem in the set of acetate-exclusive reactions since this position was occupied by lipid metabolism (Fig 2b).

We also analysed how the macrosystems were distributed in the levels of the DAGs. To this end, we computed the frequency of each macrosystem in each level of the DAG for each of the carbon sources. We found that macrosystems were not uniformly represented across levels. Instead, each macrosystem tended to be predominant in a particular region of the hierarchy. For instance, four macrosystems were only present within the first eight levels: *Nucleotide metabolism* in the first six levels, *Amino acid metabolism* and *Lipid metabolism* in the first seven levels and *Carbohydrate metabolism* in the first eight levels. Additionally, while *Cell wall biosynthesis* was also only represented in the first ten levels of the DAG, this macrosystem was predominant in levels six to eight, comprising between 52% and 82% of the reactions. Contrary to these macrosystems, *Cofactor and vitamin metabolism* was represented across all levels, with the exception of level eight. However, the latter macrosystem was markedly predominant in the last ten levels of the DAG, comprising over 86% of the reactions in the three carbon sources. Finally, *Transport* was not only the macrosystem with the largest overall representation in the DAG, but was also represented across a majority of levels, a result that reflects the utter dependence of reaction flux on import and export as well as intercellular transport processes (online Jupyter Notebook, Section 2.4).

Altogether, these findings indicate that specific metabolic systems have a preferential position in the metabolic hierarchy. Therefore, the degree to which a reaction can control the flux of others depends on the specific metabolic system to which it belongs.

## The flux order relation in experimental data

We next evaluated the extent to which molecular data profiles respected the flux order relation. First, we considered whether $^{13}$C-estimated fluxes respected the predicted flux order relation.

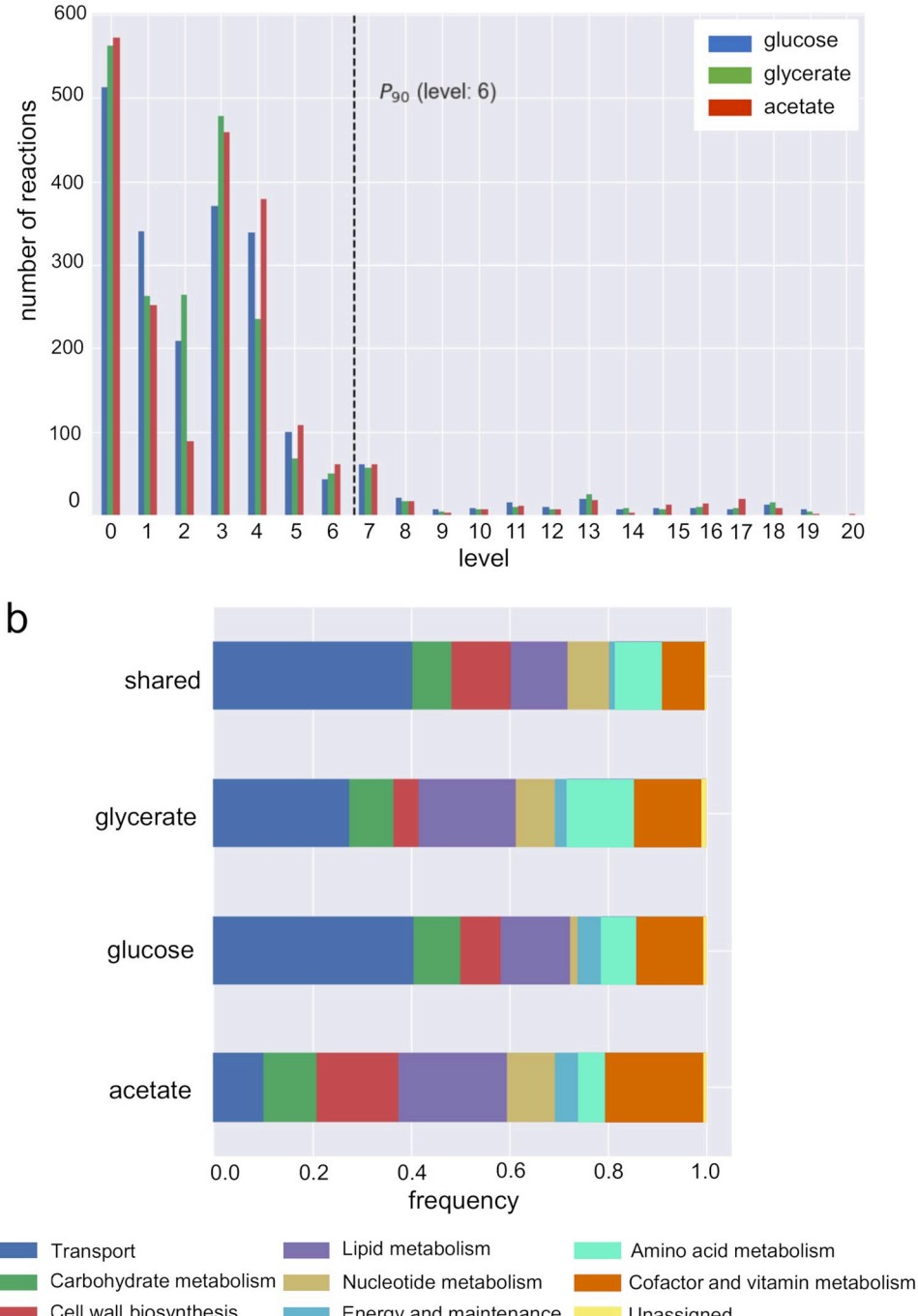

**Fig 2. Properties of the flux order directed acyclic graph (DAG).** (a) Distribution of flux-ordered pairs per graph level and carbon source, levels one to six contain approximatively 90% of the reactions in the DAG in all carbon sources. (b) Distributions of metabolic macrosystems across graph levels and carbon sources. See the online Jupyter Notebook, Section 2.4, for a complete depiction of the metabolic systems across DAG levels.

We also tested if the flux order relation was reflected in other data profiles which do not directly correspond to measurement of fluxes. Specifically, we also considered protein and transcript levels, which potentially correlate with the amount of active enzyme, as well as enzyme turnover number, i.e., $k_{cat}$ (Methods, section: Model, growth media and data preparation).

In all cases, we computed the distribution of differences of reaction-associated data values for all flux-ordered reaction pairs. In the cases where several data values were available for each reaction, i.e., flux, protein and transcript levels, we computed the average difference among all available data values for each flux-ordered reaction pair. We then employed the proportion of flux-ordered pairs with positive (averaged) data differences as our statistic to quantify the agreement of each data set with the flux order relation. Additionally, we performed a permutation analysis in which we randomized data values to evaluate the significance of our results (Methods, section: Analysis of order relations in data).

As commented in section: The flux order relation in *E. coli*, the sets of flux-ordered pairs previously analyzed were generated in an scenario of aerobic growth, in which we constrained the minimum flux through the biomass reaction to be 95% (controlled though a parameter, $\alpha$, here $\alpha = 0.95$) of the maximum possible. However, experimental measurements could correspond to a scenario where *E. coli* was growing at a larger or smaller rate. Thus, we also analyzed the flux order relation with a smaller, $\alpha = 0.925$ and a larger, $\alpha = 1$, minimum flux through the biomass reaction. As expected, the number of flux-ordered pairs decreased rapidly with $\alpha$ values smaller than 0.925 (online Jupyter Notebook, Section 6).

**The flux order relation is respected by flux data.** We found that all experimentally estimated flux values respected the flux order relation in the iJO1366 model. Specifically, all averaged mean differences were positive with an experimental *p*-value = 0.0015. However, experimental flux levels are obtained by fitting [13]C-labelling data to a metabolic network, typically considering core metabolic reactions. It is hence possible that the flux order relation observed in the flux data is imposed by the network used to estimate the fluxes and may thus represent artifacts.

To investigate this possibility, we analysed the flux order relation in the small network employed to generate most of the samples contained in the flux database employed in this study [34], here termed *experimental network* (Methods, section: Flux orders in [13]C-measured fluxes). We identified a total of 92 ordered reaction pairs in the experimental network, out of which 25 ordered pairs had available flux data. We found that, here too, flux data followed the order relation in all cases. Interestingly, the overlap between the sets of ordered reaction pairs of iJO1366 and the experimental network was small, with only five shared pairs with available data, a 20% of the ordered pairs with data in iJO1366 (online Jupyter Notebook, Section 3.1). Therefore, a majority of the flux-ordered reaction pairs followed by flux data in iJO1366 are genuine of the whole network and are not induced by the experimental network employed to estimate the flux data.

Finally, we evaluated whether flux data also followed the flux order relation when the minimum flux through the biomass reaction was constrained to be 100% of the maximum (smaller values than 95% did not render any flux-ordered pairs with available flux data). We found that the previous results did not change qualitatively. In this case, 93% of the averaged mean differences were positive, with an experimental p-value = 0.0006 (online Jupyter Notebook, Section 3.1).

**Evaluation of the order relation in transcript and protein levels.** Any enzyme has an associated metabolic cost, i.e., the amount of metabolic resources that are invested to produce one unit of enzyme. We hypothesized that, in a scenario where cells minimise costs, regulating gene expression and protein levels of enzymes to match the flux order relation of the catalysed

reactions could be economically advantageous due to avoiding overproduction of unneeded enzymes. Moreover, metabolic costs vary greatly among enzymes, which can be explained by both, the difference in protein size and also by an enrichment in costly amino acids [40]. Violation of the flux order relation could thus have a larger impact in reaction pairs catalysed by enzymes with larger metabolic costs. To test this hypothesis, we analysed to what extent transcript and protein data followed the flux order relation in the three carbon sources. In addition, we evaluated whether enzyme subgroups with increasing metabolic costs tended to better follow the flux order relation.

We first obtained publicly available enzyme costs [30] and then evaluated the flux order relation in the data for several subsets of reactions with increasing enzyme costs. Specifically, we evaluated the following subsets: (*i*) all flux-ordered reaction pairs, (*ii*) flux-ordered pairs whose enzymes had a cost at least as large as the percentile $P_{70}$, (*iii*) $P_{80}$ and (*iv*) $P_{85}$ of the distribution of all employed enzyme costs (Methods, section: Model, growth media and data preparation).

We found that the proportion of ordered reaction pairs whose transcript data matched the ordering increased across reaction groups with increasing protein costs with glucose as a carbon source (Table 2 and online Jupyter Notebook, Section 3.2). Specifically, the proportion of positive average data differences increased from 62.8%, for $P_0$, to 70.8%, for $P_{70}$, 77.2% for $P_{80}$ and 83.5% for $P_{85}$ (for all $p$-value $<5x10^{-4}$, permutation test). However, the tendency found for glucose was not observed for the other carbon sources. Specifically, while a similar result held to the one obtained for glucose at $P_0$, i.e., a proportion of positive average data differences of 62.2% and 62.2% for acetate and glycerate, respectively, none of the alternative carbon sources showed a clear increase of the proportion for larger protein costs (Table 2 and online Jupyter Notebook, Section 3.2).

We also found a tendency towards increasing the agreement between protein abundance and the flux order relation with increasing protein costs for glucose and glycerate. Specifically, the proportion of positive average differences, with glucose as sole carbon source, increased from 59.7% for $P_0$ to 62.1% for $P_{70}$, 70.7% for $P_{80}$ and 75.9% for $P_{85}$, and from 59.8% for $P_0$ to 62.6% for $P_{70}$, 72.7% for $P_{80}$ and 77.9% for $P_{85}$ with glycerate as a carbon source ($p$-values $<$ 0.04, permutation test, Table 2 and online Jupyter Notebook, Section 3.2). Similar to transcript data, protein abundance with acetate as a carbon source did not agree well with the flux order relation, and only the first case, at $P_0$, with a proportion of 57.6% positive average differences was statistically significant ($p$-value $<$ 0.03, permutation test, Table 2 and online Jupyter Notebook, Section 3.2). Therefore, we found that the agreement between data profiles and the flux order relation increased for reaction pairs with higher protein costs, although this trend was

**Table 2. The flux order relation in experimental data.** Fraction of positive average differences among flux-ordered pairs across the five data types employed, i.e. flux, transcript and protein levels and enzyme $k_{cat}$ values, for the three carbon sources and with a minimum biomass production of 95% of the maximum (i.e., $\alpha$ = 0.95). In the case of data on transcript and protein levels, results are shown for the four different protein cost percentile values, $P_0$—$P_{85}$ (see main text). Since $k_{cat}$ values can be considered constant under different carbon sources, here we employ the same dataset for the three carbon sources. Experimental $p$-values from the permutation test are displayed within parentheses. Interactive figures displaying the actual distributions of average data differences are displayed in the online Jupyter Notebook, Section 3.

| | Fluxes | Transcript | | | | Protein | | | | $k_{cat}$ |
|---|---|---|---|---|---|---|---|---|---|---|
| | | $P_0$ | $P_{70}$ | $P_{80}$ | $P_{85}$ | $P_0$ | $P_{70}$ | $P_{80}$ | $P_{85}$ | |
| Glucose | 1 (0.002) | 0.628 $(10^{-4})$ | 0.708 $(10^{-4})$ | 0.772 $(10^{-4})$ | 0.835 $(10^{-4})$ | 0.597 (0.012) | 0.621 (0.04) | 0.707 (0.03) | 0.759 (0.022) | 0.792 $(10^{-4})$ |
| Glycerate | - | 0.626 $(10^{-4})$ | 0.624 (0.032) | 0.665 (0.014) | 0.663 (0.05) | 0.598 (0.008) | 0.626 (0.048) | 0.727 (0.014) | 0.779 (0.014) | 0.79.5 $(10^{-4})$ |
| Acetate | - | 0.622 $(10^{-4})$ | 0.654 (0.002) | 0.621 (0.07) | 0.635 (0.068) | 0.576 (0.036) | 0.549 (0.194) | 0.581 (0.138) | 0.648 (0.098) | 0.79.8 $(10^{-4})$ |

only clear for transcript level data with glucose as a carbon source and for protein abundance data with glucose and glycerate as a carbon source.

Finally, we evaluated both transcript levels and protein abundance under different minimum flux through the biomass reaction. In the case of transcript levels with $\alpha = 0.925$ and glucose as carbon source the proportion of positive average differences increased from 62% at $P_0$ to 72% at $P_{80}$ ($p$-values $< 0.02$, permutation test, online Jupyter Notebook, Section 3.2), while the other carbon sources did not render significant results. When setting $\alpha = 1$, we found significant fractions of positive average differences among transcript levels only at $P_0$ in the three carbon sources, 65%, 63% and 64% for glucose, glycerate and acetate (online Jupyter Notebook, Section 3.2). In the case of protein abundances, we found proportions of positive average differences above 84% for the three carbon sources with $\alpha = 0.925$ and $\alpha = 1$, but only when protein costs were at least as large as the percentiles $P_{80}$ and $P_{85}$ (online Jupyter Notebook, Section 3.2).

**Evaluation of the order relation in enzyme $k_{cat}$ values.** We next investigated if the turnover numbers, i.e. $k_{cat}$ values, followed the flux order relation. $k_{cat}$ values are a fundamental, structural property of enzymes and, thus, independent of carbon source. Therefore, we evaluated the same dataset on the sets of ordered reaction pairs generated under the three carbon sources (Methods, Section Model, growth media and data preparation). Here again, we found that data tended to follow the predicted flux order relation. Specifically, the proportion of positive average differences was 79.2%, 79.5% and 79.8% for glucose, glycerate, and acetate, respectively (Table 2 and online Jupyter Notebook, Section 3.3, $p$-values $= 10^{-4}$, permutation test). Additionally, $k_{cat}$ values followed the same trend for $\alpha = 0.925$ and $\alpha = 1$, with fractions of positive average differences above 76% for the three carbon sources (online Jupyter Notebook, Section 3.3).

The previous results indicate that $k_{cat}$ values are the data type with the second-best agreement with the flux order relation, dominated only by $^{13}$C-based flux estimations. Additionally, since $k_{cat}$ depends on enzyme structure, our results suggest optimisation of enzyme structure throughout evolution to match the ordering pattern imposed by the metabolic network at steady state. Further, since $k_{cat}$ values are measured under substrate saturating conditions [28], our results align with *in vivo* saturating conditions for a subset of enzymes, as already observed for the central carbon metabolism of *E. coli* [41].

## Effect of the level in the flux order DAG on the number of gene regulatory interactions

As previously discussed, reactions situated in the first levels of the flux order DAG impose an upper bound to the fluxes of a large number of reactions in the lower levels. Therefore, regulating the flux through these reactions can have a greater impact in controlling the overall flux distribution. A similar concept was explored by Hosseini et al. [16], where the DAG representing the directional coupling relation in a metabolic model of *E. coli* was explored with respect to the distribution of regulatory interactions at each level. This analysis found that genes associated to reactions situated in the first levels were more likely to be transcriptionally regulated.

We conducted a similar analysis with the flux order DAG of *E. coli*, although this time we evaluated to what extent genes of reactions in prior levels exhibited more regulatory interactions than those in later levels. First, we computed the total number of regulatory interactions of the genes associated to each reaction with available gene regulatory data, which we obtained from the RegulonDB database [35]. Next, similar to Hosseini et al. [16], we partitioned the DAG levels into three sets—first, middle, last—of levels which had a similar number of reactions and compared the distributions of total regulatory interactions between the sets (Mann-

Whitney U test, $\alpha = 0.05$). Finally, we also analysed the flux coupling DAG of the iJO1366 model using the RegulonDB database (Methods, section: Gene regulation in the flux order and flux coupling DAGs).

We found that in both, the flux order and the flux coupling DAG, genes associated to reactions in the first and middle levels were significantly more regulated than those in the last level (*p*-values < 0.04, Fig 3 and online Jupyter Notebook, Section 3.4). Additionally, we obtained identical results when we constrained the flux through the biomass reaction to be the maximum possible, i.e., with $\alpha = 1$. In the case of $\alpha = 0.925$, only genes of reactions in the middle levels were significantly more regulated than those in the last levels (online Jupyter Notebook, Section 6). Therefore, these results support the hypothesis that genes associated to reactions in higher positions, in both DAGs, are more regulated due to their greater control capability.

### Flux order relation and reaction essentiality

Identifying essential reactions—reactions that must carry non-zero flux to enable cellular growth—is crucial in both biotechnological and biomedical settings. For instance, in the first case, essential reactions and their genes form a set that cannot be removed during metabolic pathway optimisation approaches. In the second case, essential reactions and their enzymes provide potential targets to halt the growth of pathogens [42–44].

All reactions that are flux-ordered with the biomass (pseudo)reaction are also essential, due to the upper flux bound imposed on biomass production. However, it is unclear whether all essential reactions are also flux-ordered with the biomass reaction. To clarify this issue, we next investigated to what extent essential reactions were also flux-ordered with the biomass reaction (Methods, section: The flux order relation in the context of reaction essentiality). We found a total of 27 reactions that were both, essential and flux-ordered with the biomass reaction under growth on glucose, i.e., were predecessors of the biomass reaction in the flux order DAG (Fig 4 and online Jupyter Notebook, Section 4). Flux-ordered and essential reactions were organized in four levels in which four metabolic macrosystems were represented, in decreasing order of the number of reactions these were: *Transport*, *Carbohydrate metabolism*, *Amino acid metabolism* and *Energy and maintenance*. However, we found that most essential reactions (a total of 347) were not flux-ordered with the biomass reaction, demonstrating that reactions do not need to be flux-ordered with the biomass reaction to be essential (online Jupyter Notebook, Section 4). We conclude that the flux order relation can be used to identify other, subtle limitations to the biomass reaction flux, i.e., reactions that continuously constraint biomass production as opposed to the binary switch imposed by the unordered essential reactions. The latter can be readily used in relevant biotechnological applications, such as control and fine-tuning of growth-limiting genes.

### Conclusion

Understanding hierarchies in biological networks not only can contribute to the discovery of underlying design principles, but also provide the means for effective control of these complex networks. Hierarchies in biological networks are multifaceted, and thus two principle approaches have been used to formalize and uncover them, namely, hierarchies of embedded subnetworks and hierarchies of individual components of the network.

Since metabolic networks are composed of metabolites interconverted by biochemical reactions, the hierarchies following the second approach can be based on ordering of either metabolites of reactions. Here, we wanted to explore the extent to which the structure of the metabolic network alone imposes a hierarchy of reactions based on ordering of their fluxes. Such a view provides the means to discover a hierarchy that is present irrespective of the

a

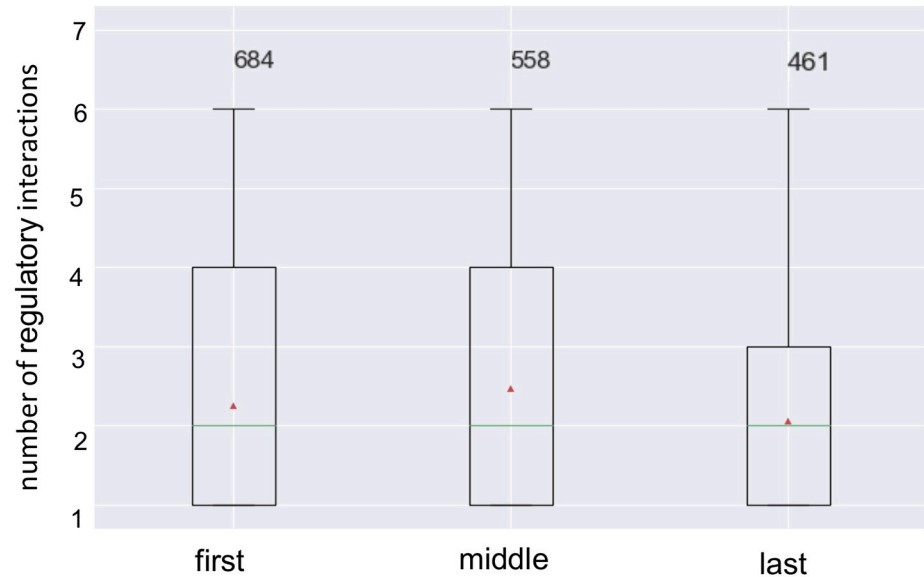

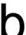

b

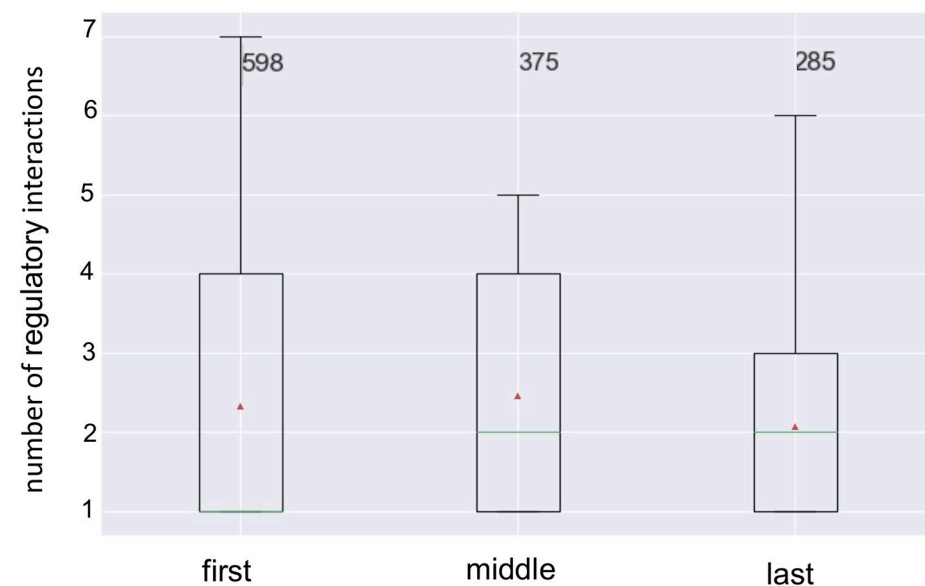

**Fig 3. Evaluation of the flux order relation in *E. coli*'s gene regulatory network.** Relationship between the number of gene regulatory interactions of the enzyme-coding genes and the position in the hierarchy of the corresponding reactions. (a) The flux order DAG levels have been grouped into three categories: those occupying the first, middle and last levels (see main text). The distribution of the total number of gene regulatory interactions (both positive and negative) is represented for the three categories as box plots. The green line represents the median number of interactions while the red triangle represents the mean. (b) A similar analysis employing the directional flux coupling DAG. In both cases, the first and the middle and the middle and the last levels contain significantly larger numbers of gene regulatory interactions (p-value < 0.04, see main text).

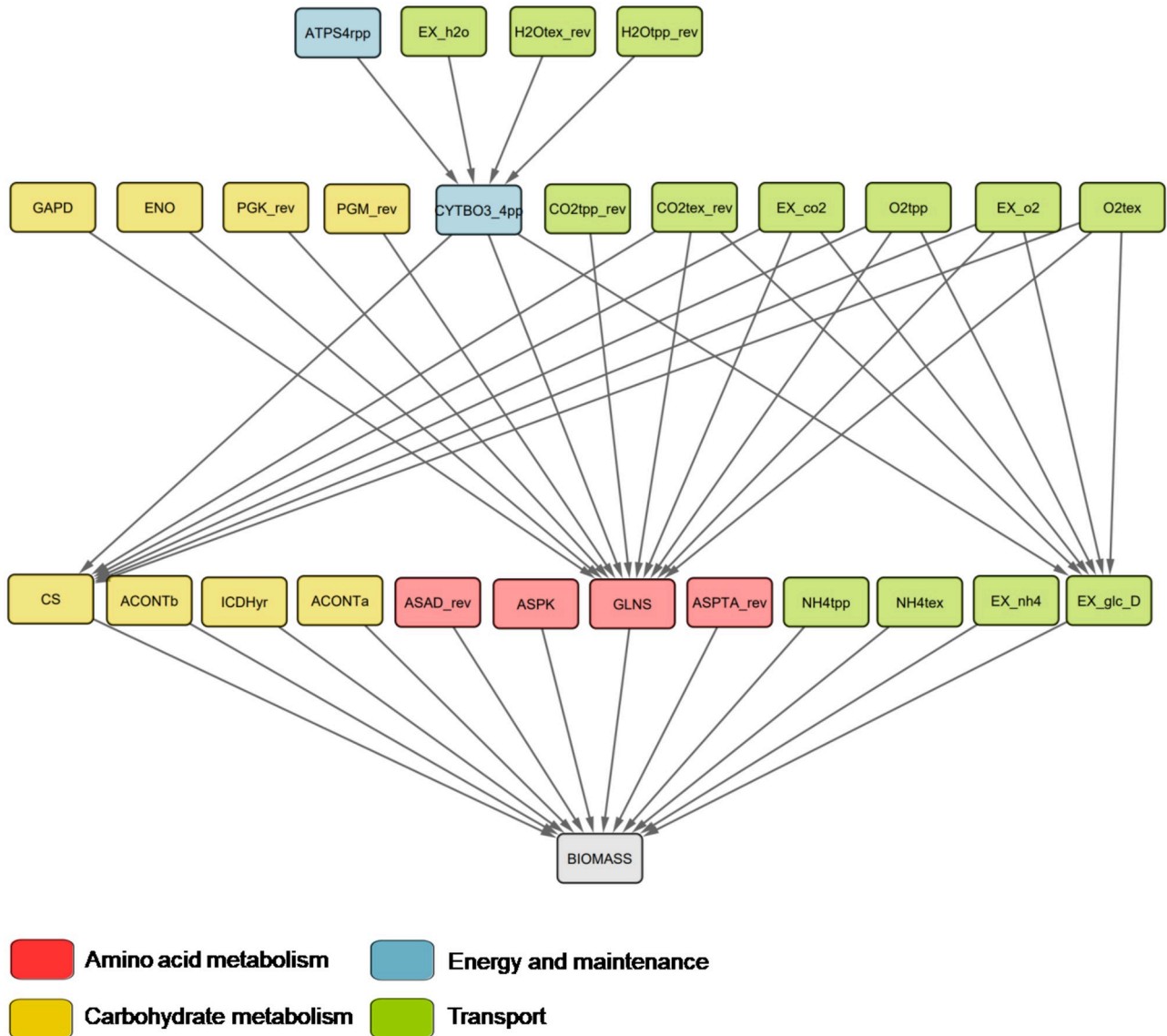

**Fig 4. Subgraph of the flux order DAG containing all predecessors of the biomass reaction for growth under glucose.** All reactions represented in this DAG carry greater or equal fluxes than the biomass reaction. Hence, they are all essential reactions since they would impose an upper flux bound of zero to the biomass reaction if they were to be inactive. Only four reaction macrosystems are represented in the set of reactions: Amino acid metabolism, carbohydrate metabolism, Energy and maintenance and Transport, with transport being the one with the largest number of reactions. ATPS4rpp: ATP synthase, EX_h2o: $H_2O$ exchange, H2Otex_rev: $H_2O$ transport periplasm to extracellular, H2Otpp_rev: $H_2O$ transport cytoplasm to periplasm, GAPD: Glyceraldehyde-3-phosphate dehydrogenase, ENO: Enolase, PGK_rev: Phosphoglycerate kinase (reverse), PGM_rev: Phosphoglycerate mutase (reverse), CYTBO3_4pp: Cytochrome oxidase, CO2tpp_rev: $CO_2$ transporter cytoplasm to periplasm, CO2tex_rev: $CO_2$ transport periplasm to extracellular, EX_co2: $CO_2$ exchange, O2tpp: $O_2$ transport periplasm to cytoplasm, EX_o2: $O_2$ exchange, O2tex: $O_2$ transport extracellular to periplasm, CS: Citrate synthase, ACONTb: Aconitase, ICDHyr: Isocitrate dehydrogenase, ACONTa: Aconitase, ASAD_rev: Aspartate-semialdehyde dehydrogenase (reverse), ASPK: Aspartate kinase, GLNS: Glutamine synthetase, ASPTA_rev: Aspartate transaminase (reverse), NH4tpp: Ammonia transport periplasm to cytoplasm, NH4tex: Ammonia transport extracellular to periplasm, EX_nh4: Ammonia exchange, EX_glc_D: Glucose import.

particularities of the reaction kinetics, for which there is still limited information at genome-wide scale. Moreover, this view can be readily cast in the constraint-based modelling framework and can contribute to the discovery of important functional limits of metabolic networks. The latter is due to the natural definition of the flux ordering relation, whereby reactions

further up the hierarchy dominate reactions down the hierarchy with respect to their flux in every steady state supported by the network.

Our formalization of the flux order relation facilitated the discovery of a hierarchy of reactions in the metabolic network reconstruction of the bacterium *E. coli*. Analysis of the graph representation pointed out that the hierarchy is branched and relatively shallow for three carbon sources. In addition, we found a complete agreement between the flux order relation and [13]C-based flux profiles. Interestingly, we also found a partial agreement when analysing other phenotypic profiles which correlate with reaction flux, i.e., transcript and protein levels, as well as enzyme catalytic constants. In the first case, these findings point at optimisation of resource allocation to minimise costs of enzyme synthesis. This hypothesis that was further supported by an increased agreement found in subsets of enzymes with increasing costs, particularly when glucose was the carbon source, in transcript and protein levels, and when glycerate was the carbon source, in protein levels. In the second case, the findings suggest network constraints acting upon the evolution of enzyme structural properties to match the flux order relation.

The concept of flux order relation in metabolic networks can be readily extended to analyse the hierarchy outside of steady state dynamics, provided we have insights in the degree of metabolic concentration changes around a steady state [45, 46]. In addition, due to the universality of a large part of metabolism, it is plausible that the flux order relation also holds in metabolic networks of other organisms. It would be particularly interesting to evaluate to what extent the hierarchy holds in eukaryotic organisms, in which metabolism is compartmentalized and more complex. Further, one can expand the concept for applications to metabolic networks with given analytically tractable kinetics, e.g. mass action [47]. Finally, since the flux order relation identifies reactions imposing a fine-tuned control over a target reaction, the applications of the hierarchy in metabolic engineering remain as a promising direction for future research.

## Supporting information

**S1 Table. Metabolic subsystems and macrosystems of iJO1366.** All subsystems contained in the iJO1366 model have been assigned to one of eight metabolic macrosystems: Amino acid metabolism, carbohydrate metabolism, energy and maintenance metabolism, cell wall biosynthesis, cofactor and vitamin metabolism, lipid metabolism, nucleotide metabolism and transport.
(XLSX)

**S2 Table. Stoichiometric matrix of the experimental network used to estimate fluxes.** The stoichiometric data employed to generate the network has been obtained from supplementary files SFigure2 and S4- and S5 tables from Ishii et al. 2007 (reference [34]). Reactions r2, r7 appear as reversible in Supplementary Tables S4-5 but irreversible in Figure S2 of the original publication [34]. Here, we take them as reversible. Additionally, some reactions in iJO1366 are reversible while they are classified as irreversible in the original publication [34]. For consistency, these reactions contain the tag "_reverse" to match the ids in iJO1366.
(XLSX)

## Author Contributions

**Conceptualization:** Semidán Robaina-Estévez, Zoran Nikoloski.

**Data curation:** Semidán Robaina-Estévez.

**Formal analysis:** Semidán Robaina-Estévez.

**Funding acquisition:** Zoran Nikoloski.

**Investigation:** Semidán Robaina-Estévez.

**Methodology:** Semidán Robaina-Estévez.

**Software:** Semidán Robaina-Estévez.

**Supervision:** Zoran Nikoloski.

**Validation:** Semidán Robaina-Estévez.

**Visualization:** Semidán Robaina-Estévez.

**Writing – original draft:** Semidán Robaina-Estévez, Zoran Nikoloski.

**Writing – review & editing:** Semidán Robaina-Estévez, Zoran Nikoloski.

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
