## [Decision Letter · Decision Letter 0]

29 Oct 2019

Dear Dr Robaina Estévez,

Thank you very much for submitting your manuscript 'Flux-based hierarchical organization of Escherichia coli's metabolic network' for review by PLOS Computational Biology. Your manuscript has been fully evaluated by the PLOS Computational Biology editorial team and in this case also by independent peer reviewers. The reviewers appreciated the attention to an important problem, but raised some substantial concerns about the manuscript as it currently stands. While your manuscript cannot be accepted in its present form, we are willing to consider a revised version in which the issues raised by the reviewers have been adequately addressed. Please especially consider the points raised by the reviewers with regard to prior work investigating hierarchical organization of the metabolic network. We cannot, of course, promise publication at that time.

Sincerely,

Christoph Kaleta

Associate Editor

PLOS Computational Biology

Jason Papin

Editor-in-Chief

PLOS Computational Biology

[LINK]

Reviewer's Responses to Questions

**Comments to the Authors:**

Reviewer #1: The authors investigate the broad field that biological networks across scales exhibit hierarchical organization that may constrain network function. They find that metabolic networks show a hierarchy of reactions based on a natural flux ordering that holds for every steady state, even reflected in experimental measurements of transcript, protein and flux levels of Escherichia coli under various growth conditions as well as in the catalytic rate constants of the corresponding enzymes.

This is nice and well documented here both by the formal analysis as well as the public experimental data used for validation of these findings and a nice analysis.

Comments:

--What should be discussed somewhat more is the direct implications of the results and possible explanations:

Certainly, there is some input from resource partitioning and there is fine-tuning of enzyme levels in E. coli and of course it has to respect the constraints imposed by the network structure at steady state.

However, all this may even more be the result of other more general factors, in particular the evolution of enzyme networks or just the result of fitness optimization, in particular as E.coli is metabolically so active and broad.

--I agree that reactions in upper layers of the hierarchy impose an upper bound on the flux of the reactions downstream, however, it would be good if the authors can show a nice concrete example on how the hierarchical organization of metabolism due to the flux ordering has direct applications in metabolic engineering, for instance in plants.

Again, there is no doubt that you can do metabolic engineering by constraint-based modelling, but it would be good if the authors can clarify how then the insight in the natural flux ordering presents then a handle to do something novel which is not already obvious from e.g. calculating the flux modes according to standard techniques.

Reviewer #2: In their manuscript "flux-based hierarchical organization ..." the authors propose a new method for computing a hierarchy among metabolic reactions.

Employing the framework of steady-state fluxes obtained via LP from genome-scale metabolic models, the criterion for ranking reactions is flux ordering. The authors compare the resulting hierarchy with categories derived from metabolic pathways, with flux measurements and with regulatory information.

I find the manuscript suitable for PLOS CB. However, I have a few concerns and comments, which the authors should address.

(1) From my perspective the literature around the hierarchical organization of metabolic systems (and biological systems in general) should be cited on a broader level. For example in [Yu, H., & Gerstein, M. (2006). Genomic analysis of the hierarchical structure of regulatory networks. Proceedings of the National Academy of Sciences, 103(40), 14724-14731] the distribution of gene attributes across hierarchical levels has been studied (which is also the strategy employed in the present manuscript). Also, as the authors point out in the introduction, the hierarchical organization of metabolic systems can be assessed on the level of metabolites, as well as on the level of reactions. Examples for the former, which the authors may consider to cite are: [Matthäus, F., Salazar, C., & Ebenhöh, O. (2008). Biosynthetic potentials of metabolites and their hierarchical organization. PLoS Computational Biology, 4(4), e1000049], [Matthäus, F., Salazar, C., & Ebenhöh, O. (2008). Biosynthetic potentials of metabolites and their hierarchical organization. PLoS Computational Biology, 4(4), e1000049]. Regarding the comparison with transcriptional regulation, I regard the findings from [Shlomi, T., Eisenberg, Y., Sharan, R., & Ruppin, E. (2007). A genome‐scale computational study of the interplay between transcriptional regulation and metabolism. Molecular systems biology, 3(1)] to be highly relevant. The authors should compare their results with the 'determined' and 'non-determined' categories from this paper.

(2) I was surprised that the authors do not discuss the fact that metabolic fluxes follow a power-law distribution [Almaas, E., Kovacs, B., Vicsek, T., Oltvai, Z. N., & Barabási, A. L. (2004). Global organization of metabolic fluxes in the bacterium Escherichia coli. Nature, 427(6977), 839-843]. Doesn't this already imply a hierarchical organization?

(3) How do the authors motivate their choice of iJO1366 as the genome-scale metabolic model? Wouldn't the more recent model iML1515 from [Monk, J. M., Lloyd, C. J., Brunk, E., Mih, N., Sastry, A., King, Z., ... & Feist, A. M. (2017). iML1515, a knowledgebase that computes Escherichia coli traits. Nature biotechnology, 35(10), 904] been a more plausible choice?

(4) In Section 2.2 the authors study the distribution of pathways across the hierarchical levels. Maybe I missed this point, but shouldn't these percentages be compared to a null model of shuffled macrosystem labels, in order to eliminate (level and system) size effects?

(5) I have not understood the argument in Section 2.5.1 regarding the potential artifcact due to the empirical focus of flux data on the metabolic core. Wouldn't the appropriate test for this be to evaluate the flux order graph obtained from the E. coli 'metabolic core model' from [Orth, J. D., Fleming, R. M., & Palsson, B. O. (2010). Reconstruction and use of microbial metabolic networks: the core Escherichia coli metabolic model as an educational guide. EcoSal plus]?

(6) Given the scope of comparisons in Sections 2.5.2, I would recommend a correction for multiple testing for the p-values reported here.

(7) I might have misunderstood the analysis strategy of the authors here, but when they say 'a reaction is connected to another by a directional edge if it carries a greater of equal flux in any steady state', do they mean 'any steady state compatible with the minimum flux through the biomass reaction' or do they literally mean any vertex of the flux polytope? I suppose it is the former. In this case, however, the results will depend strongly on the choice of this minimal flux to be 95 percent of the maximally feasible biomass production rate (as stated in Section 4.7. Should this be the case, I strongly recommend to study the key results under variation of this parameter.

**Have all data underlying the figures and results presented in the manuscript been provided?**

Reviewer #1: Yes

Reviewer #2: Yes

PLOS authors have the option to publish the peer review history of their article (what does this mean?). If published, this will include your full peer review and any attached files.

Reviewer #1: No

Reviewer #2: No

---

## [Decision Letter · Decision Letter 1]

19 Mar 2020

Dear Dr. Robaina Estévez,

Thank you very much for submitting th revision of your manuscript "Flux-based hierarchical organization of Escherichia coli's metabolic network" for consideration at PLOS Computational Biology. As you can see the reviewers appreciated your revision but reviewer 1 still had one point which I like you to address in a minor revision of the manuscript.

Sincerely,

Christoph Kaleta

Associate Editor

PLOS Computational Biology

Jason Papin

Editor-in-Chief

PLOS Computational Biology

[LINK]

Reviewer's Responses to Questions

**Comments to the Authors:**

Reviewer #1: The authors revised their manuscript thoroughly, one question remains:

"as discussed in Section 2.7, the difference is that essential reactions which are also flux-ordered affect flux through the biomass reaction in a continuous manner, while essential reactions which are not flux-ordered affect biomass production in a binary (on/off upon a threshold value) manner. Hence, identifying essential reactions which are flux-ordered with the biomass reaction could allow fine tuning of reaction flux to control biomass production."

This is fair enough, but please discuss why not identifying the essential reactions just by looking at those modes which drop out if an enzyme is inhibited does not do a similar good job (if not at least one mode drops out by inhibition of the enzyme this is not an essential enzyme). I would even think that this method is simpler and allows easy to quantify the more important enzymes (as blocking these inhibits a higher number of modes).

--So as mentioned already regarding the previous version of the manuscript, please make still a bit clearer, best by looking at this remaining comment, why and where your approach is preferable to classical elementary mode analysis and what you really gain by using this more complex method of yours.

Reviewer #2: The revised version of the manuscript "Flux-based hierarchical organization of Escherichia coli's metabolic network" (PCOMPBIOL-D-19-01496R1) is, from my perspective, a substantial improvement.

I agree with their assessment that the results of Shlomi et al. (2007) cannot be directly compared with their findings.

Regarding the work by Almaas et al. (2004), I believe that the manuscript would benefit from a short comment on power-law flux distributions (as most readers will expect this in a publication titled "Flux-based hierarchical organization ...").

Regarding the choice of the E. coli model, I found the authors' response to my question perfectly convincing.

I still disagree with the authors' statement that "empirical p-values" do not require a correction for multiple testing. If I consider all groups of flux-ordered reaction pairs assessed here as an individual hypothesis, I would argue that a large set of hypotheses has been tested (and hence a multiple-testing correction would be required). However, this decision is the responsibility of the authors (they explain quite clearly what they are doing).

The point about the dependence of the results on the choice of the minimal flux has been addressed properly by the authors, as have been the other (mostly minor) remarks I had.

Summarizing, I have no objection against acceptance of the manuscript (leaving it up to the authors to address the point about the power-law flux distribution or the different opinion about multiple testing, should they decide to do so).

**Have all data underlying the figures and results presented in the manuscript been provided?**

Reviewer #1: Yes

Reviewer #2: Yes

PLOS authors have the option to publish the peer review history of their article (what does this mean?). If published, this will include your full peer review and any attached files.

Reviewer #1: No

Reviewer #2: No
---

## [Decision Letter · Decision Letter 2]

30 Mar 2020

Dear Dr. Robaina Estévez,

We are pleased to inform you that your manuscript 'Flux-based hierarchical organization of Escherichia coli's metabolic network' has been provisionally accepted for publication in PLOS Computational Biology.

Best regards,

Christoph Kaleta

Associate Editor

PLOS Computational Biology

Jason Papin

Editor-in-Chief

PLOS Computational Biology

Reviewer's Responses to Questions

**Comments to the Authors:**

Reviewer #1: fine

**Have all data underlying the figures and results presented in the manuscript been provided?**

Reviewer #1: Yes

PLOS authors have the option to publish the peer review history of their article (what does this mean?). If published, this will include your full peer review and any attached files.

Reviewer #1: No

---

## [Editor Report · Acceptance letter]

10 Apr 2020

PCOMPBIOL-D-19-01496R2 

Flux-based hierarchical organization of Escherichia coli's metabolic network

Dear Dr Robaina Estévez,

I am pleased to inform you that your manuscript has been formally accepted for publication in PLOS Computational Biology. Your manuscript is now with our production department and you will be notified of the publication date in due course.

With kind regards,

Sarah Hammond
